# Robust and interpretable blind image denoising via bias-free convolutional neural networks

**Sreyas Mohan**[*]
Center for Data Science
New York University
sm7582@nyu.edu

**Zahra Kadkhodaie**[*]
Center for Data Science
New York University
zk388@nyu.edu

**Eero P. Simoncelli**
Center for Neural Science, and
Howard Hughes Medical Institute
New York University
eero.simoncelli@nyu.edu

**Carlos Fernandez-Granda**
Center for Data Science, and
Courant Inst. of Mathematical Sciences
New York University
cfgranda@cims.nyu.edu

## Abstract

We study the generalization properties of deep convolutional neural networks for image denoising in the presence of varying noise levels. We provide extensive empirical evidence that current state-of-the-art architectures systematically overfit to the noise levels in the training set, performing very poorly at new noise levels. We show that strong generalization can be achieved through a simple architectural modification: removing all additive constants. The resulting "bias-free" networks attain state-of-the-art performance over a broad range of noise levels, even when trained over a narrow range. They are also locally linear, which enables direct analysis with linear-algebraic tools. We show that the denoising map can be visualized locally as a filter that adapts to both image structure and noise level. In addition, our analysis reveals that deep networks implicitly perform a projection onto an adaptively-selected low-dimensional subspace, with dimensionality inversely proportional to noise level, that captures features of natural images.

## 1 Introduction and Contributions

The problem of denoising consists of recovering a signal from measurements corrupted by noise, and is a canonical application of statistical estimation that has been studied since the 1950's. Achieving high-quality denoising results requires (at least implicitly) quantifying and exploiting the differences between signals and noise. In the case of photographic images, the denoising problem is both an important application, as well as a useful test-bed for our understanding of natural images. In the past decade, convolutional neural networks (LeCun et al., 2015) have achieved state-of-the-art results in image denoising (Zhang et al., 2017; Chen & Pock, 2017). Despite their success, these solutions are mysterious: we lack both intuition and formal understanding of the mechanisms they implement. Network architecture and functional units are often borrowed from the image-recognition literature, and it is unclear which of these aspects contributes to, or limits, the denoising performance. The goal of this work is advance our understanding of deep-learning models for denoising. Our contributions are twofold: First, we study the generalization capabilities of deep-learning models across different noise levels. Second, we provide novel tools for analyzing the mechanisms implemented by neural networks to denoise natural images.

An important advantage of deep-learning techniques over traditional methodology is that a single neural network can be trained to perform denoising at a wide range of noise levels. Currently, this is achieved by simulating the whole range of noise levels during training (Zhang et al., 2017). Here, we show that this is not necessary. Neural networks can be made to *generalize automatically across noise*

---

[*]Equal contribution.

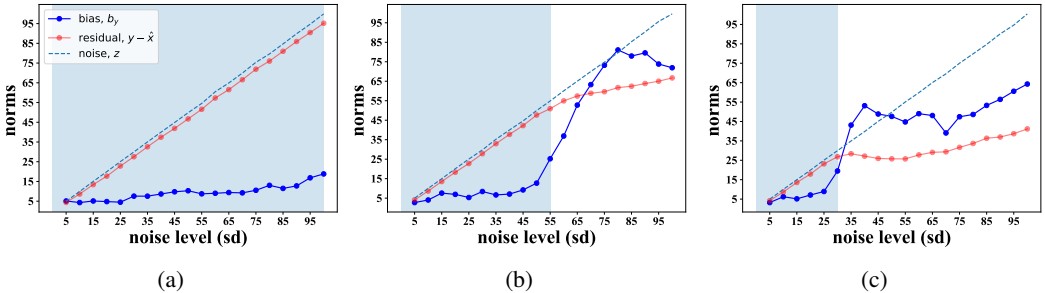

Figure 1: First-order analysis of the residual of a denoising convolutional neural network as a function of noise level. The plots show the norms of the residual and the net bias averaged over 100 $20 \times 20$ natural-image patches for networks trained over different training ranges. The range of noises used for training is highlighted in blue. **(a)** When the network is trained over the full range of noise levels ($\sigma \in [0, 100]$) the net bias is small, growing slightly as the noise increases. **(b-c)** When the network is trained over the a smaller range ($\sigma \in [0, 55]$ and $\sigma \in [0, 30]$), the net bias grows explosively for noise levels beyond the training range. This coincides with a dramatic drop in performance, reflected in the difference between the magnitudes of the residual and the true noise. The CNN used for this example is DnCNN (Zhang et al., 2017); using alternative architectures yields similar results as shown in Figure 8.

*levels* through a simple modification in the architecture: removing all additive constants. We find this holds for a variety of network architectures proposed in previous literature. We provide extensive empirical evidence that the main state-of-the-art denoising architectures systematically overfit to the noise levels in the training set, and that this is due to the presence of a net bias. Suppressing this bias makes it possible to attain state-of-the-art performance while training over a very limited range of noise levels.

The data-driven mechanisms implemented by deep neural networks to perform denoising are *almost completely unknown*. It is unclear what priors are being learned by the models, and how they are affected by the choice of architecture and training strategies. Here, we provide novel linear-algebraic tools to visualize and interpret these strategies through a local analysis of the Jacobian of the denoising map. The analysis reveals locally adaptive properties of the learned models, akin to existing nonlinear filtering algorithms. In addition, we show that the deep networks implicitly perform a projection onto an adaptively-selected low-dimensional subspace capturing features of natural images.

## 2 RELATED WORK

The classical solution to the denoising problem is the Wiener filter (Wiener, 1950), which assumes a translation-invariant Gaussian signal model. The main limitation of Wiener filtering is that it over-smoothes, eliminating fine-scale details and textures. Modern filtering approaches address this issue by adapting the filters to the local structure of the noisy image (e.g. Tomasi & Manduchi (1998); Milanfar (2012)). Here we show that neural networks implement such strategies implicitly, learning them directly from the data.

In the 1990's powerful denoising techniques were developed based on multi-scale ("wavelet") transforms. These transforms map natural images to a domain where they have sparser representations. This makes it possible to perform denoising by applying nonlinear thresholding operations in order to discard components that are small relative to the noise level (Donoho & Johnstone, 1995; Simoncelli & Adelson, 1996; Chang et al., 2000). From a linear-algebraic perspective, these algorithms operate by projecting the noisy input onto a lower-dimensional subspace that contains plausible signal content. The projection eliminates the orthogonal complement of the subspace, which mostly contains noise. This general methodology laid the foundations for the state-of-the-art models in the 2000's (e.g. (Dabov et al., 2006)), some of which added a data-driven perspective, learning sparsifying transforms (Elad & Aharon, 2006), and nonlinear shrinkage functions (Hel-Or & Shaked, 2008; Raphan & Simoncelli, 2008), directly from natural images. Here, we show that deep-learning models learn similar priors in the form of local linear subspaces capturing image features.

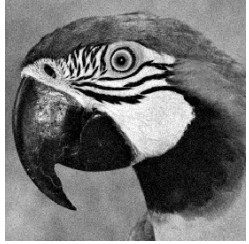
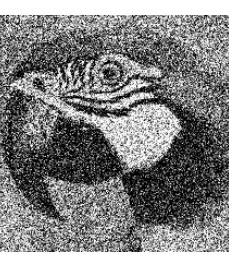
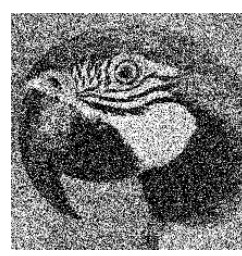
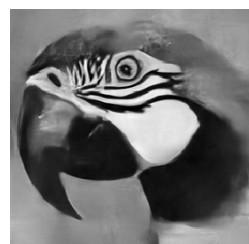

| Noisy training image, | Noisy test image, | Test image, denoised | Test image, denoised |
| $\sigma = 10$ (max level) | $\sigma = 90$ | by CNN | by BF-CNN |

Figure 2: Denoising of an example natural image by a CNN and its bias-free counterpart (BF-CNN), both trained over noise levels in the range $\sigma \in [0, 10]$ (image intensities are in the range $[0, 255]$). The CNN performs poorly at high noise levels ($\sigma = 90$, far beyond the training range), whereas BF-CNN performs at state-of-the-art levels. The CNN used for this example is DnCNN (Zhang et al., 2017); using alternative architectures yields similar results (see Section 5).

In the past decade, purely data-driven models based on convolutional neural networks (LeCun et al., 2015) have come to dominate all previous methods in terms of performance. These models consist of cascades of convolutional filters, and rectifying nonlinearities, which are capable of representing a diverse and powerful set of functions. Training such architectures to minimize mean square error over large databases of noisy natural-image patches achieves current state-of-the-art results (Zhang et al., 2017; Huang et al., 2017; Ronneberger et al., 2015; Zhang et al., 2018a).

## 3 NETWORK BIAS IMPAIRS GENERALIZATION

We assume a measurement model in which images are corrupted by additive noise: $y = x + n$, where $x \in \mathbb{R}^N$ is the original image, containing $N$ pixels, $n$ is an image of i.i.d. samples of Gaussian noise with variance $\sigma^2$, and $y$ is the noisy observation. The denoising problem consists of finding a function $f : \mathbb{R}^N \to \mathbb{R}^N$, that provides a good estimate of the original image, $x$. Commonly, one minimizes the mean squared error : $f = \arg\min_g E||x - g(y)||^2$, where the expectation is taken over some distribution over images, $x$, as well as over the distribution of noise realizations. In deep learning, the denoising function $g$ is parameterized by the weights of the network, so the optimization is over these parameters. If the noise standard deviation, $\sigma$, is unknown, the expectation must also be taken over a distribution of $\sigma$. This problem is often called *blind denoising* in the literature. In this work, we study the generalization performance of CNNs *across* noise levels $\sigma$, i.e. when they are tested on noise levels not included in the training set.

Feedforward neural networks with rectified linear units (ReLUs) are piecewise affine: for a given activation pattern of the ReLUs, the effect of the network on the input is a cascade of linear transformations (convolutional or fully connected layers, $W_k$), additive constants ($b_k$), and pointwise multiplications by a binary mask corresponding to the fixed activation pattern ($R$). Since each of these is affine, the entire cascade implements a single affine transformation. For a fixed noisy input image $y \in \mathbb{R}^N$ with $N$ pixels, the function $f : \mathbb{R}^N \to \mathbb{R}^N$ computed by a denoising neural network may be written

$$f(y) = W_L R(W_{L-1}...R(W_1 y + b_1) + ...b_{L-1}) + b_L = A_y y + b_y, \tag{1}$$

where $A_y \in \mathbb{R}^{N \times N}$ is the Jacobian of $f(\cdot)$ evaluated at input $y$, and $b_y \in \mathbb{R}^N$ represents the *net bias*. The subscripts on $A_y$ and $b_y$ serve as a reminder that both depend on the ReLU activation patterns, which in turn depend on the input vector $y$.

Based on equation 1 we can perform a first-order decomposition of the error or *residual* of the neural network for a specific input: $y - f(y) = (I - A_y)y - b_y$. Figure 1 shows the magnitude of the residual and the constant, which is equal to the net bias $b_y$, for a range of noise levels. Over the training range, the net bias is small, implying that the linear term is responsible for most of the denoising (see Figures 9 and 10 for a visualization of both components). However, when the network is evaluated at noise levels outside of the training range, the norm of the bias increases dramatically, and the residual is significantly smaller than the noise, suggesting a form of overfitting. Indeed, network performance

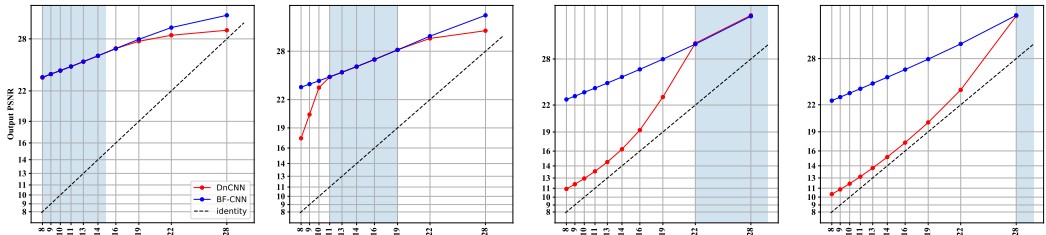

Figure 3: Comparison of the performance of a CNN and a BF-CNN with the same architecture for the experimental design described in Section 5. The performance is quantified by the PSNR of the denoised image as a function of the input PSNR. Both networks are trained over a fixed ranges of noise levels indicated by a blue background. In all cases, the performance of BF-CNN generalizes robustly beyond the training range, while that of the CNN degrades significantly. The CNN used for this example is DnCNN (Zhang et al., 2017); using alternative architectures yields similar results (see Figures 11 and 12).

generalizes very poorly to noise levels outside the training range. This is illustrated for an example image in Figure 2, and demonstrated through extensive experiments in Section 5.

## 4 PROPOSED METHODOLOGY: BIAS-FREE NETWORKS

Section 3 shows that CNNs overfit to the noise levels present in the training set, and that this is associated with wild fluctuations of the net bias $b_y$. This suggests that the overfitting might be ameliorated by removing additive (bias) terms from every stage of the network, resulting in a *bias-free* CNN (BF-CNN). Note that bias terms are also removed from the batch-normalization used during training. This simple change in the architecture has an interesting consequence. If the CNN has ReLU activations the denoising map is locally homogeneous, and consequently *invariant to scaling*: rescaling the input by a constant value simply rescales the output by the same amount, just as it would for a linear system.

**Lemma 1.** *Let $f_{\mathrm{BF}} : \mathbb{R}^N \to \mathbb{R}^N$ be a feedforward neural network with ReLU activation functions and no additive constant terms in any layer. For any input $y \in \mathbb{R}$ and any nonnegative constant $\alpha$,*

$$f_{\mathrm{BF}}(\alpha y) = \alpha f_{\mathrm{BF}}(y). \tag{2}$$

*Proof.* We can write the action of a bias-free neural network with $L$ layers in terms of the weight matrix $W_i$, $1 \le i \le L$, of each layer and a rectifying operator $\mathcal{R}$, which sets to zero any negative entries in its input. Multiplying by a nonnegative constant does not change the sign of the entries of a vector, so for any $z$ with the right dimension and any $\alpha > 0$ $\mathcal{R}(\alpha z) = \alpha \mathcal{R}(z)$, which implies

$$f_{\mathrm{BF}}(\alpha y) = W_L \mathcal{R}(W_{L-1} \cdots \mathcal{R}(W_1 \alpha y)) = \alpha W_L \mathcal{R}(W_{L-1} \cdots \mathcal{R}(W_1 y)) = \alpha f_{\mathrm{BF}}(y). \tag{3}$$

$\square$

Note that networks with nonzero net bias are not scaling invariant because scaling the input may change the activation pattern of the ReLUs. Scaling invariance is intuitively desireable for a denoising method operating on natural images; a rescaled image is still an image. Note that Lemma 1 holds for networks with skip connections where the feature maps are concatenated or added, because both of these operations are linear.

In the following sections we demonstrate that removing all additive terms in CNN architectures has two important consequences: (1) the networks gain the ability to generalize to noise levels not encountered during training (as illustrated by Figure 2 the improvement is striking), and (2) the denoising mechanism can be analyzed locally via linear-algebraic tools that reveal intriguing ties to more traditional denoising methodology such as nonlinear filtering and sparsity-based techniques.

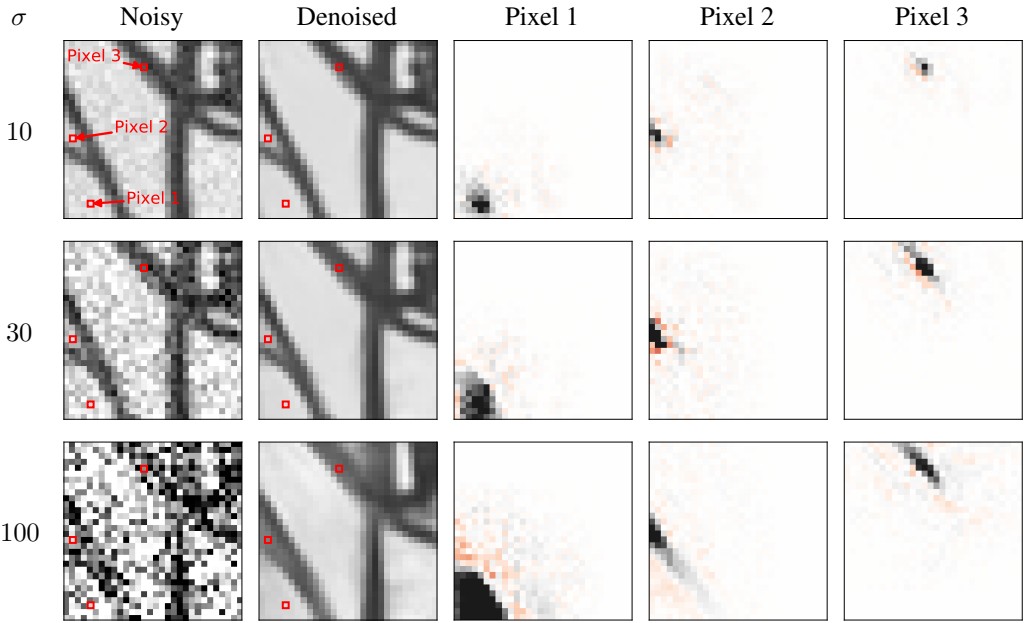

Figure 4: Visualization of the linear weighting functions (rows of $A_y$ in equation 4) of a BF-CNN for three example pixels of an input image, and three levels of noise. The images in the three rightmost columns show the weighting functions used to compute each of the indicated pixels (red squares). All weighting functions sum to one, and thus compute a local average (note that some weights are negative, indicated in red). Their shapes vary substantially, and are adapted to the underlying image content. As the noise level $\sigma$ increases, the spatial extent of the weight functions increases in order to average out the noise, while respecting boundaries between different regions in the image, which results in dramatically different functions for each pixel. The CNN used for this example is DnCNN (Zhang et al., 2017); using alternative architectures yields similar results (see Figure 13).

## 5   BIAS-FREE NETWORKS GENERALIZE ACROSS NOISE LEVELS

In order to evaluate the effect of removing the net bias in denoising CNNs, we compare several state-of-the-art architectures to their bias-free counterparts, which are exactly the same except for the absence of any additive constants within the networks (note that this includes the batch-normalization additive parameter). These architectures include popular features of existing neural-network techniques in image processing: recurrence, multiscale filters, and skip connections. More specifically, we examine the following models (see Section A for additional details):

- DnCNN (Zhang et al., 2017): A feedforward CNN with $20$ convolutional layers, each consisting of $3 \times 3$ filters, $64$ channels, batch normalization (Ioffe & Szegedy, 2015), a ReLU nonlinearity, and a skip connection from the initial layer to the final layer.

- Recurrent CNN: A recurrent architecture inspired by Zhang et al. (2018a) where the basic module is a CNN with 5 layers, $3 \times 3$ filters and $64$ channels in the intermediate layers. The order of the recurrence is 4.

- UNet (Ronneberger et al., 2015): A multiscale architecture with 9 convolutional layers and skip connections between the different scales.

- Simplified DenseNet: CNN with skip connections inspired by the DenseNet architecture (Huang et al., 2017; Zhang et al., 2018b).

We train each network to denoise images corrupted by i.i.d. Gaussian noise over a range of standard deviations (the *training range* of the network). We then evaluate the network for noise levels that are both within and beyond the training range. Our experiments are carried out on $180 \times 180$ natural images from the Berkeley Segmentation Dataset (Martin et al., 2001) to be consistent with previous

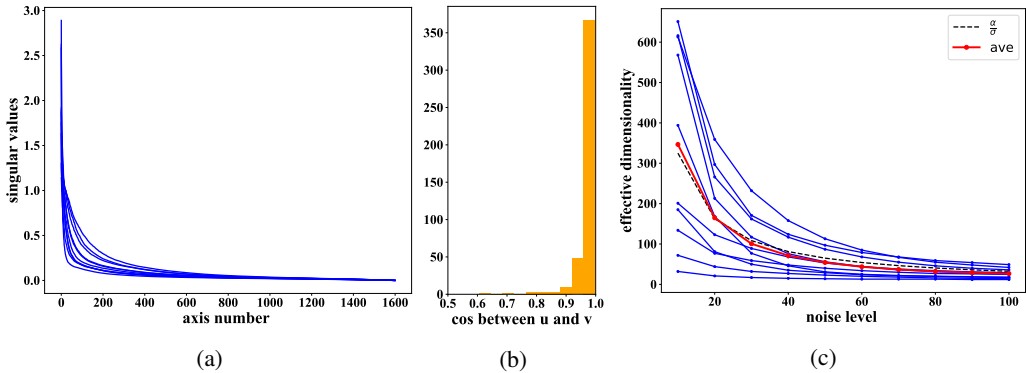

Figure 5: Analysis of the SVD of the Jacobian of a BF-CNN for ten natural images, corrupted by noise of standard deviation $\sigma = 50$. **(a)** Singular value distributions. For all images, a large proportion of the values are near zero, indicating (approximately) a projection onto a subspace (the *signal subspace*). **(b)** Histogram of dot products (cosine of angle) between the left and right singular vectors that lie within the signal subspaces. **(c)** Effective dimensionality of the signal subspaces (computed as sum of squared singular values) as a function of noise level. For comparison, the total dimensionality of the space is 1600 ($40 \times 40$ pixels). Average dimensionality (red curve) falls approximately as the inverse of $\sigma$ (dashed curve). The CNN used for this example is DnCNN (Zhang et al., 2017); using alternative architectures yields similar results (see Figure 17).

results (Schmidt & Roth, 2014; Chen & Pock, 2017; Zhang et al., 2017). Additional details about the dataset and training procedure are provided in Section B.

Figures 3, 11 and 12 show our results. For a wide range of different training ranges, and for all architectures, we observe the same phenomenon: the performance of CNNs is good over the training range, but degrades dramatically at new noise levels; in stark contrast, the corresponding BF-CNNs provide strong denoising performance over noise levels outside the training range. This holds for both PSNR and the more perceptually-meaningful Structural Similarity Index (Wang et al., 2004) (see Figure 12). Figure 2 shows an example image, demonstrating visually the striking difference in generalization performance between a CNN and its corresponding BF-CNN. Our results provide strong evidence that removing net bias in CNN architectures results in effective generalization to noise levels out of the training range.

## 6 Revealing the Denoising Mechanisms Learned by BF-CNNs

In this section we perform a local analysis of BF-CNN networks, which reveals the underlying denoising mechanisms learned from the data. A bias-free network is strictly linear, and its net action can be expressed as

$$f_{\text{BF}}(y) = W_L R(W_{L-1}...R(W_1 y)) = A_y y, \tag{4}$$

where $A_y$ is the Jacobian of $f_{\text{BF}}(\cdot)$ evaluated at $y$. The Jacobian at a fixed input provides a local characterization of the denoising map. In order to study the map we perform a linear-algebraic analysis of the Jacobian. Our approach is similar in spirit to visualization approaches– proposed in the context of image classification– that differentiate neural-network functions with respect to their input (e.g. Simonyan et al. (2013); Montavon et al. (2017)).

### 6.1 Nonlinear adaptive filtering

The linear representation of the denoising map given by equation 4 implies that the $i$th pixel of the output image is computed as an inner product between the $i$th row of $A_y$, denoted $a_y(i)$, and the input image:

$$f_{\text{BF}}(y)(i) = \sum_{j=1}^{N} A_y(i,j)y(j) = a_y(i)^T y. \tag{5}$$

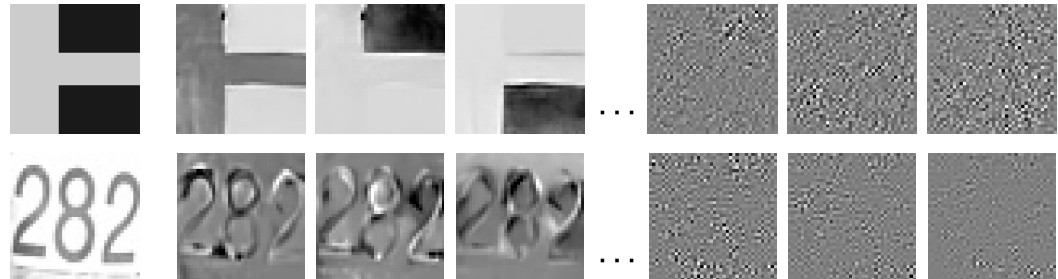

Figure 6: Visualization of left singular vectors of the Jacobian of a BF-CNN, evaluated on two different images (top and bottom rows), corrupted by noise with standard deviation $\sigma = 50$. The left column shows original (clean) images. The next three columns show singular vectors corresponding to non-negligible singular values. The vectors capture features from the clean image. The last three columns on the right show singular vectors corresponding to singular values that are almost equal to zero. These vectors are noisy and unstructured. The CNN used for this example is DnCNN (Zhang et al., 2017); using alternative architectures yields similar results (see Figure 16).

The vectors $a_y(i)$ can be interpreted as *adaptive filters* that produce an estimate of the denoised pixel via a weighted average of noisy pixels. Examination of these filters reveals their diversity, and their relationship to the underlying image content: they are adapted to the local features of the noisy image, averaging over homogeneous regions of the image without blurring across edges. This is shown for two separate examples and a range of noise levels in Figures 4, 13, 14 and 15 for the architectures described in Section 5. We observe that the equivalent filters of all architectures adapt to image structure.

Classical Wiener filtering (Wiener, 1950) denoises images by computing a local average dependent on the noise level. As the noise level increases, the averaging is carried out over a larger region. As illustrated by Figures 4, 13, 14 and 15, the equivalent filters of BF-CNNs also display this behavior. The crucial difference is that the filters are adaptive. The BF-CNNs learn such filters implicitly from the data, in the spirit of modern nonlinear spatially-varying filtering techniques designed to preserve fine-scale details such as edges (e.g. Tomasi & Manduchi (1998), see also Milanfar (2012) for a comprehensive review, and Choi et al. (2018) for a recent learning-based approach).

## 6.2 PROJECTION ONTO ADAPTIVE LOW-DIMENSIONAL SUBSPACES

The local linear structure of a BF-CNN facilitates analysis of its functional capabilities via the singular value decomposition (SVD). For a given input $y$, we compute the SVD of the Jacobian matrix: $A_y = USV^T$, with $U$ and $V$ orthogonal matrices, and $S$ a diagonal matrix. We can decompose the effect of the network on its input in terms of the left singular vectors $\{U_1, U_2 \ldots, U_N\}$ (columns of $U$), the singular values $\{s_1, s_2 \ldots, s_N\}$ (diagonal elements of $S$), and the right singular vectors $\{V_1, V_2, \ldots V_N\}$ (columns of $V$):

$$f_{\text{BF}}(y) = A_y y = USV^T y = \sum_{i=1}^{N} s_i (V_i^T y) U_i. \tag{6}$$

The output is a linear combination of the left singular vectors, each weighted by the projection of the input onto the corresponding right singular vector, and scaled by the corresponding singular value.

Analyzing the SVD of a BF-CNN on a set of ten natural images reveals that most singular values are very close to zero (Figure 5a). The network is thus discarding all but a very low-dimensional portion of the input image. We also observe that the left and right singular vectors corresponding to the singular values with non-negligible amplitudes are approximately the same (Figure 5b). This means that the Jacobian is (approximately) symmetric, and we can interpret the action of the network as projecting the noisy signal onto a low-dimensional subspace, as is done in wavelet thresholding schemes. This is confirmed by visualizing the singular vectors as images (Figure 6). The singular vectors corresponding to non-negligible singular values are seen to capture features of the input image; those corresponding to near-zero singular values are unstructured. The BF-CNN therefore implements

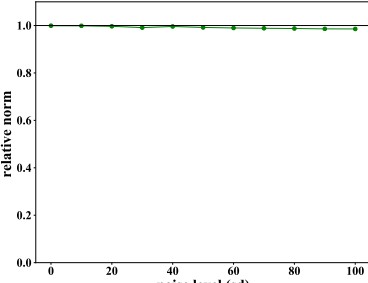 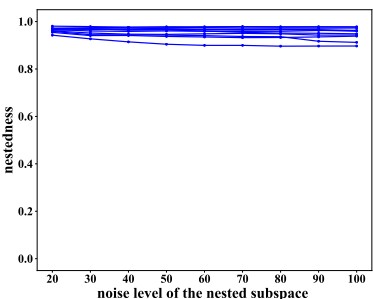

Figure 7: Signal subspace properties. **Left:** Signal subspace, computed from Jacobian of a BF-CNN evaluated at a particular noise level, contains the clean image. Specifically, the fraction of squared $\ell_2$ norm preserved by projection onto the subspace is nearly one as $\sigma$ grows from 10 to 100 (relative to the image pixels, which lie in the range $[0, 255]$). Results are averaged over 50 example clean images. **Right:** Signal subspaces at different noise levels are nested. The subspace axes for a higher noise level lie largely within the subspace obtained for the lowest noise level ($\sigma = 10$), as measured by the sum of squares of their projected norms. Results are shown for 10 example clean images.

an approximate projection onto an adaptive *signal subspace* that preserves image structure, while suppressing the noise.

We can define an "effective dimensionality" of the signal subspace as $d := \sum_{i=1}^{N} s_i^2$, the amount of variance captured by applying the linear map to an $N$-dimensional Gaussian noise vector with variance $\sigma^2$, normalized by the noise variance. The remaining variance equals

$$E_n||A_y n||^2 = E_n||U_y S_y V_y^T n||^2 = E_n||S_y n||^2 = E_n \sum_{i=1}^{N} s_i^2 n_i^2 = \sum_{i=1}^{N} s_i^2 E_n(n_i^2) \approx \sigma^2 \sum_{i=1}^{N} s_i^2,$$

where $E_n$ indicates expectation over noise $n$, so that $d = E_n||A_y n||^2/\sigma^2 = \sum_{i=1}^{N} s_i^2$.

When we examine the preserved signal subspace, we find that the clean image lies almost completely within it. For inputs of the form $y := x + n$ (where $x$ is the clean image and $n$ the noise), we find that the subspace spanned by the singular vectors up to dimension $d$ contains $x$ almost entirely, in the sense that projecting $x$ onto the subspace preserves most of its energy. This holds for the whole range of noise levels over which the network is trained (Figure 7).

We also find that for any given clean image, the effective dimensionality of the signal subspace ($d$) decreases systematically with noise level (Figure 5c). At lower noise levels the network detects a richer set of image features, and constructs a larger signal subspace to capture and preserve them. Empirically, we found that (on average) $d$ is approximately proportional to $\frac{1}{\sigma}$ (see dashed line in Figure 5c). These signal subspaces are nested: the subspaces corresponding to lower noise levels contain more than 95% of the subspace axes corresponding to higher noise levels (Figure 7).

Finally, we note that this behavior of the signal subspace dimensionality, combined with the fact that it contains the clean image, explains the observed denoising performance across different noise levels (Figure 3). Specifically, if we assume $d \approx \alpha/\sigma$, the mean squared error is proportional to $\sigma$:

$$\begin{aligned}
\text{MSE} &= E_n||A_y(x + n) - x||^2 \\
&\approx E_n||A_y n||^2 \\
&\approx \sigma^2 d \\
&\approx \alpha \, \sigma
\end{aligned} \tag{7}$$

Note that this result runs contrary to the intuitive expectation that MSE should be proportional to the noise variance, which would be the case if the denoiser operated by projecting onto a fixed subspace. The scaling of MSE with the square root of the noise variance implies that the PSNR of the denoised image should be a linear function of the input PSNR, with a slope of $1/2$, consistent with the empirical results shown in Figure 3. Note that this behavior holds even when the networks are trained only on modest levels of noise (e.g., $\sigma \in [0, 10]$).

## 7 DISCUSSION

In this work, we show that removing constant terms from CNN architectures ensures strong generalization across noise levels, and also provides interpretability of the denoising method via linear-algebra techniques. We provide insights into the relationship between bias and generalization through a set of observations. Theoretically, we argue that if the denoising network operates by projecting the noisy observation onto a linear space of "clean" images, then that space should include all rescalings of those images, and thus, the origin. This property can be guaranteed by eliminating bias from the network. Empirically, in networks that allow bias, the net bias of the trained network is quite small within the training range. However, outside the training range the net bias grows dramatically resulting in poor performance, which suggests that the bias may be the cause of the failure to generalize. In addition, when we remove bias from the architecture, we preserve performance within the training range, but achieve near-perfect generalization, even to noise levels more than 10x those in the training range. These observations do not fully elucidate how our network achieves its remarkable generalization- only that bias prevents that generalization, and its removal allows it.

It is of interest to examine whether bias removal can facilitate generalization in noise distributions beyond Gaussian, as well as other image-processing tasks, such as image restoration and image compression. We have trained bias-free networks on uniform noise and found that they generalize outside the training range. In fact, bias-free networks trained for Gaussian noise generalize well when tested on uniform noise (Figures 18 and 19). In addition, we have applied our methodology to image restoration (simultaneous deblurring and denoising). Preliminary results indicate that bias-free networks generalize across noise levels for a fixed blur level, whereas networks with bias do not (Figure 20). An interesting question for future research is whether it is possible to achieve generalization across blur levels. Our initial results indicate that removing bias is not sufficient to achieve this.

Finally, our linear-algebraic analysis uncovers interesting aspects of the denoising map, but these interpretations are very local: small changes in the input image change the activation patterns of the network, resulting in a change in the corresponding linear mapping. Extending the analysis to reveal global characteristics of the neural-network functionality is a challenging direction for future research.

## ACKNOWLEDGEMENTS

This work was partially supported by the Howard Hughes Medical Institute (HHMI).

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

# A DESCRIPTION OF DENOISING ARCHITECTURES

In this section we describe the denoising architectures used for our computational experiments in more detail.

## A.1 DNCNN

We implement BF-DnCNN based on the architecture of the Denoising CNN (DnCNN) (Zhang et al., 2017). DnCNN consists of 20 convolutional layers, each consisting of $3 \times 3$ filters and 64 channels, batch normalization (Ioffe & Szegedy, 2015), and a ReLU nonlinearity. It has a skip connection from the initial layer to the final layer, which has no nonlinear units. To construct a bias-free DnCNN (BF-DnCNN) we remove all sources of additive bias, including the mean parameter of the batch-normalization in every layer (note however that the scaling parameter is preserved).

## A.2 RECURRENT CNN

Inspired by Zhang et al. (2018a), we consider a recurrent framework that produces a denoised image estimate of the form $\hat{x}_t = f(\hat{x}_{t-1}, y_{\text{noisy}})$, at time $t$ where $f$ is a neural network. We use a 5-layer fully convolutional network with $3 \times 3$ filters in all layers and 64 channels in each intermediate layer to implement $f$. We initialize the denoised estimate as the noisy image, i.e $\hat{x}_0 := y_{\text{noisy}}$. For the version of the network with net bias, we add trainable additive constants to every filter in all but the last layer. During training, we run the recurrence for a maximum of $T$ times, sampling $T$ uniformly at random from $\{1, 2, 3, 4\}$ for each mini-batch. At test time we fix $T = 4$.

## A.3 UNET

Our UNet model (Ronneberger et al., 2015) has the following layers:

1. *conv1* - Takes in input image and maps to 32 channels with $5 \times 5$ convolutional kernels.
2. *conv2* - Input: 32 channels. Output: 32 channels. $3 \times 3$ convolutional kernels.
3. *conv3* - Input: 32 channels. Output: 64 channels. $3 \times 3$ convolutional kernels with stride 2.
4. *conv4*- Input: 64 channels. Output: 64 channels. $3 \times 3$ convolutional kernels.
5. *conv5*- Input: 64 channels. Output: 64 channels. $3 \times 3$ convolutional kernels with dilation factor of 2.
6. *conv6*- Input: 64 channels. Output: 64 channels. $3 \times 3$ convolutional kernels with dilation factor of 4.
7. *conv7*- Transpose Convolution layer. Input: 64 channels. Output: 64 channels. $4 \times 4$ filters with stride 2.
8. *conv8*- Input: 96 channels. Output: 64 channels. $3 \times 3$ convolutional kernels. The input to this layer is the concatenation of the outputs of layer *conv7* and *conv2*.
9. *conv9*- Input: 32 channels. Output: 1 channels. $5 \times 5$ convolutional kernels.

The structure is the same as in Zhang et al. (2018a), but without recurrence. For the version with bias, we add trainable additive constants to all the layers other than *conv9*. This configuration of UNet assumes even width and height, so we remove one row or column from images in with odd height or width.

## A.4 SIMPLIFIED DENSENET

Our simplified version of the DenseNet architecture (Huang et al., 2017) has 4 blocks in total. Each block is a fully convolutional 5-layer CNN with $3 \times 3$ filters and 64 channels in the intermediate layers with ReLU nonlinearity. The first three blocks have an output layer with 64 channels while the last block has an output layer with only one channel. The output of the $i^{th}$ block is concatenated with the input noisy image and then fed to the $(i + 1)^{th}$ block, so the last three blocks have 65 input channels. In the version of the network with bias, we add trainable additive parameters to all the layers except for the last layer in the final block.

## B    Datasets and training procedure

Our experiments are carried out on $180 \times 180$ natural images from the Berkeley Segmentation Dataset (Martin et al., 2001). We use a training set of $400$ images. The training set is augmented via downsampling, random flips, and random rotations of patches in these images (Zhang et al., 2017). A test set containing $68$ images is used for evaluation. We train the DnCNN and it's bias free model on patches of size $50 \times 50$, which yields a total of 541,600 clean training patches. For the remaining architectures, we use patches of size $128 \times 128$ for a total of 22,400 training patches.

We train DnCNN and its bias-free counterpart using the Adam Optimizer (Kingma & Ba, 2014) over $70$ epochs with an initial learning rate of $10^{-3}$ and a decay factor of $0.5$ at the $50^{th}$ and $60^{th}$ epochs, with no early stopping. We train the other models using the Adam optimizer with an initial learning rate of $10^{-3}$ and train for 50 epochs with a learning rate schedule which decreases by a factor of $0.25$ if the validation PSNR decreases from one epoch to the next. We use early stopping and select the model with the best validation PSNR.

## C    Additional results

In this section we report additional results of our computational experiments:

- Figure 8 shows the first-order analysis of the residual of the different architectures described in Section A, except for DnCNN which is shown in Figure 1.
- Figures 9 and 10 visualize the linear and net bias terms in the first-order decomposition of an example image at different noise levels.
- Figure 11 shows the PSNR results for the experiments described in Section 5.
- Figure 12 shows the SSIM results for the experiments described in Section 5.
- Figures 13, 14 and 15 show the equivalent filters at several pixels of two example images for different architectures (see Section 6.1).
- Figure 16 shows the singular vectors of the Jacobian of different BF-CNNs (see Section 6.2).
- Figure 17 shows the singular values of the Jacobian of different BF-CNNs (see Section 6.2).
- Figure 18 and 19 shows that networks trained on noise samples drawn from Gaussian distribution with $0$ mean generalizes to noise drawn from uniform distribution with $0$ mean during test time. Experiments follow the procedure described in Section 5 except that the networks are evaluated on a different noise distribution during the test time.
- Figure 20 shows the application of BF-CNN and CNN to the task of image restoration, where the image is corrupted with both noise and blur at the same time. We show that BF-CNNs can generalize outside the training range for noise levels for a fixed blur level, but do not outperform CNN when generalizing to unseen blur levels.

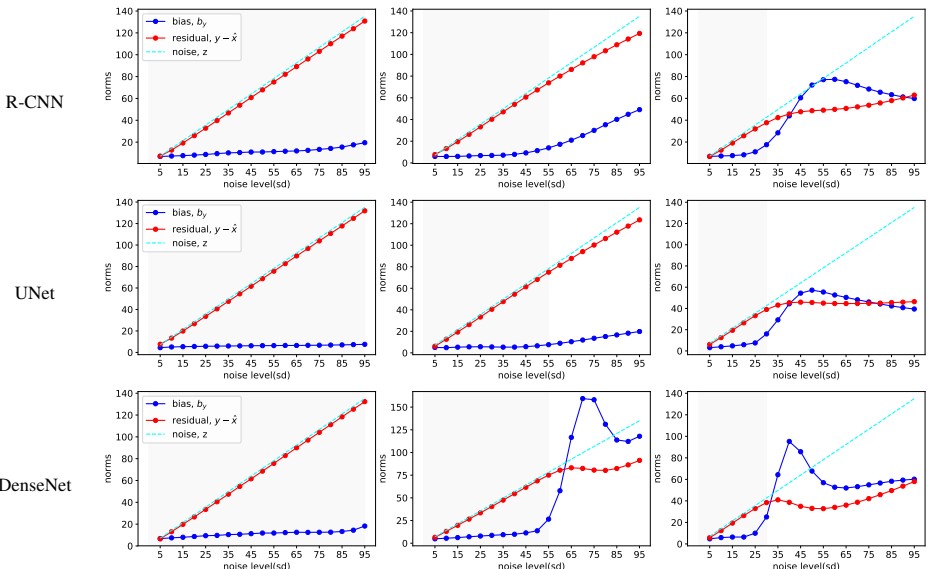

Figure 8: First-order analysis of the residual of Recurrent-CNN (Section A.2), UNet (Section A.3) and DenseNet (Section A.4) as a function of noise level. The plots show the magnitudes of the residual and the net bias averaged over 68 images in Set68 test set of Berkeley Segmentation Dataset (Martin et al., 2001) for networks trained over different training ranges. The range of noises used for training is highlighted in gray. (left) When the network is trained over the full range of noise levels ($\sigma \in [0, 100]$) the net bias is small, growing slightly as the noise increases. (middle and right) When the network is trained over the a smaller range ($\sigma \in [0, 55]$ and $\sigma \in [0, 30]$), the net bias grows explosively for noise levels outside the training range. This coincides with the dramatic drop in performance due to overfitting, reflected in the difference between the residual and the true noise.

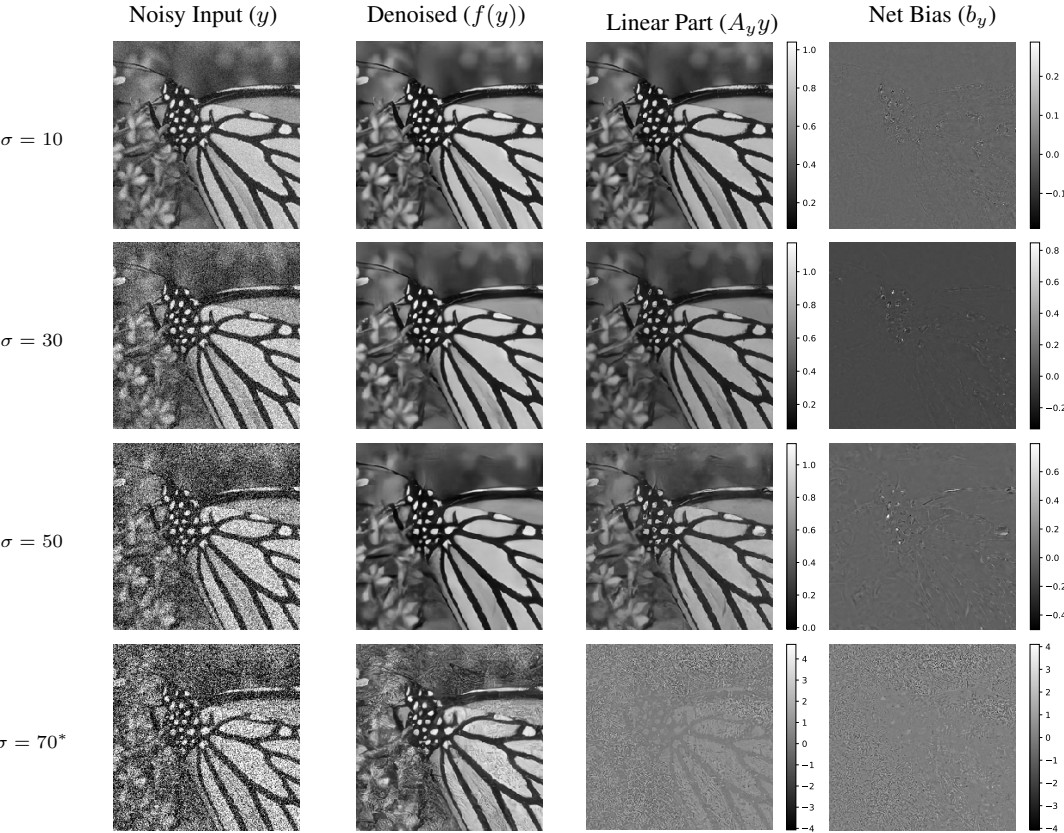

Figure 9: Visualization of the decomposition of output of DnCNN trained for noise range $[0, 55]$ into linear part and net bias. The noise level $\sigma = 70$ (highlighted by $*$) is outside the training range. Over the training range, the net bias is small, and the linear part is responsible for most of the denoising effort. However, when the network is evaluated out of the training range, the contribution of the bias increases dramatically, which coincides with a significant drop in denoising performance.

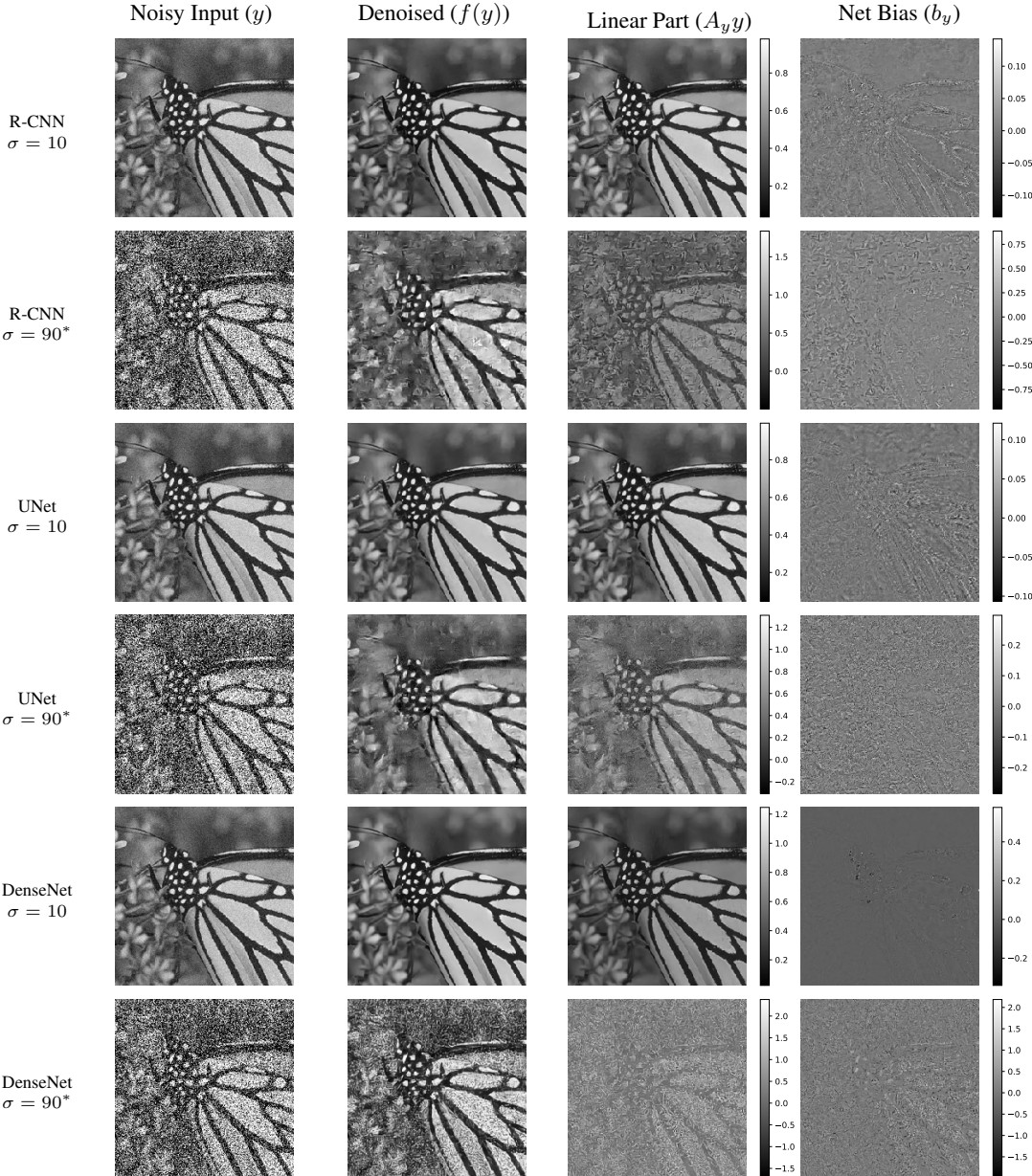

Figure 10: Visualization of the decomposition of output of Recurrent-CNN (Section A.2, UNet (Section A.3) and DenseNet (Section A.4) trained for noise range $[0, 55]$ into linear part and net bias. The noise level $\sigma = 90$ (highlighted by $*$) is outside the training range. Over the training range, the net bias is small, and the linear part is responsible for most of the denoising effort. However, when the network is evaluated out of the training range, the contribution of the bias increases dramatically, which coincides with a significant drop in denoising performance.

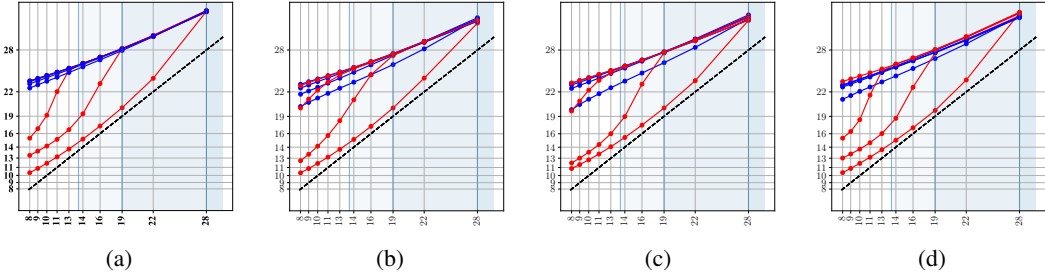

Figure 11: Comparisons of architectures with (red curves) and without (blue curves) a net bias for the experimental design described in Section 5. The performance is quantified by the PSNR of the denoised image as a function of the input PSNR of the noisy image. All the architectures with bias perform poorly out of their training range, whereas the bias-free versions all achieve excellent generalization across noise levels. **(a)** Deep Convolutional Neural Network, DnCNN (Zhang et al., 2017). **(b)** Recurrent architecture inspired by DURR (Zhang et al., 2018a). **(c)** Multiscale architecture inspired by the UNet (Ronneberger et al., 2015). **(d)** Architecture with multiple skip connections inspired by the DenseNet (Huang et al., 2017).

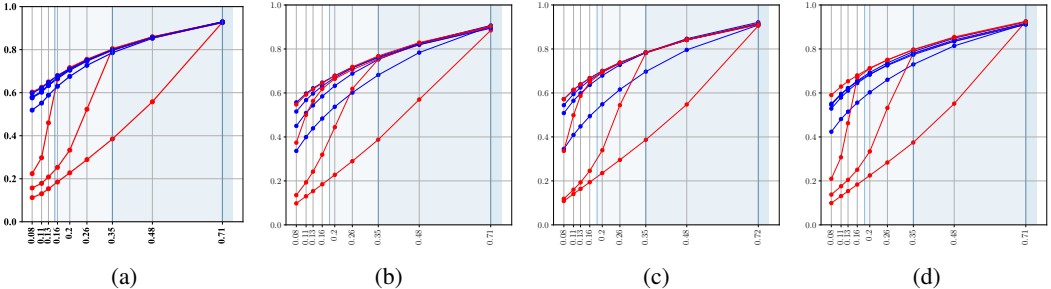

Figure 12: Comparisons of architectures with (red curves) and without (blue curves) a net bias for the experimental design described in Section 5. The performance is quantified by the SSIM of the denoised image as a function of the input SSIM of the noisy image. All the architectures with bias perform poorly out of their training range, whereas the bias-free versions all achieve excellent generalization across noise levels. **(a)** Deep Convolutional Neural Network, DnCNN (Zhang et al., 2017). **(b)** Recurrent architecture inspired by DURR (Zhang et al., 2018a). **(c)** Multiscale architecture inspired by the UNet (Ronneberger et al., 2015). **(d)** Architecture with multiple skip connections inspired by the DenseNet (Huang et al., 2017).

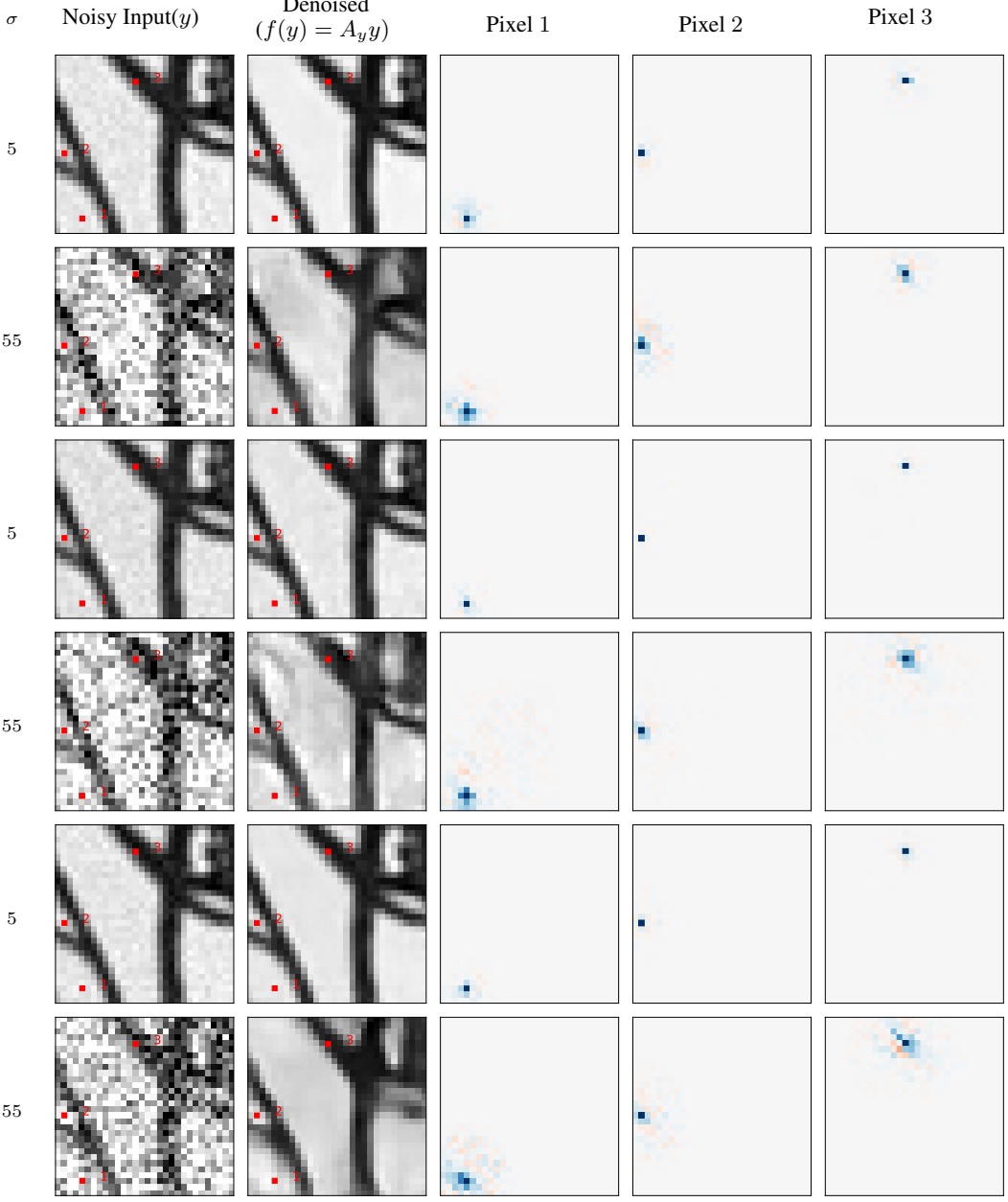

Figure 13: Visualization of the linear weighting functions (rows of $A_y$) of Bias-Free Recurrent-CNN (top 2 rows) (Section A.2), Bias-Free UNet (next 2 rows) (Section A.3) and Bias-Free DenseNet (bottom 2 rows) (Section A.4) for three example pixels of a noisy input image (left). The next image is the denoised output. The three images on the right show the linear weighting functions corresponding to each of the indicated pixels (red squares). All weighting functions sum to one, and thus compute a local average (although some weights are negative, indicated in red). Their shapes vary substantially, and are adapted to the underlying image content. Each row corresponds to a noisy input with increasing $\sigma$ and the filters adapt by averaging over a larger region.

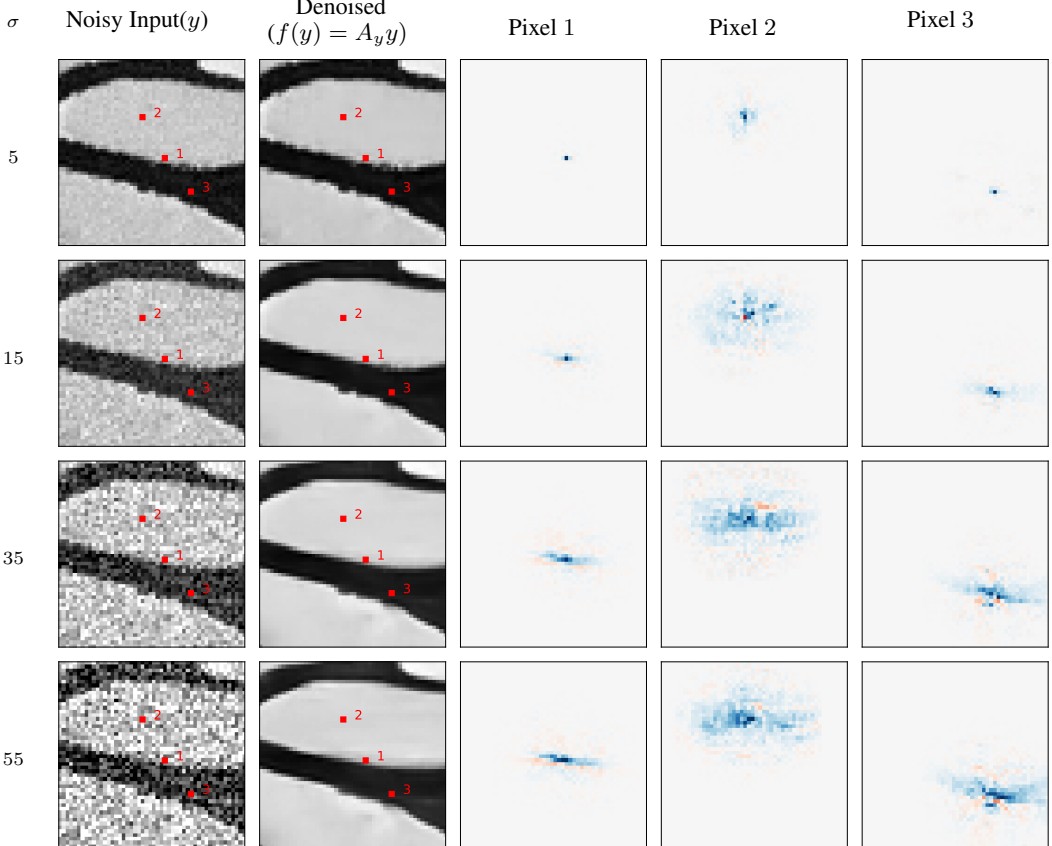

Figure 14: Visualization of the linear weighting functions (rows of $A_y$) of a BF-DnCNN for three example pixels of a noisy input image (left). The next image is the denoised output. The three images on the right show the linear weighting functions corresponding to each of the indicated pixels (red squares). All weighting functions sum to one, and thus compute a local average (although some weights are negative, indicated in red). Their shapes vary substantially, and are adapted to the underlying image content. Each row corresponds to a noisy input with increasing $\sigma$ and the filters adapt by averaging over a larger region.

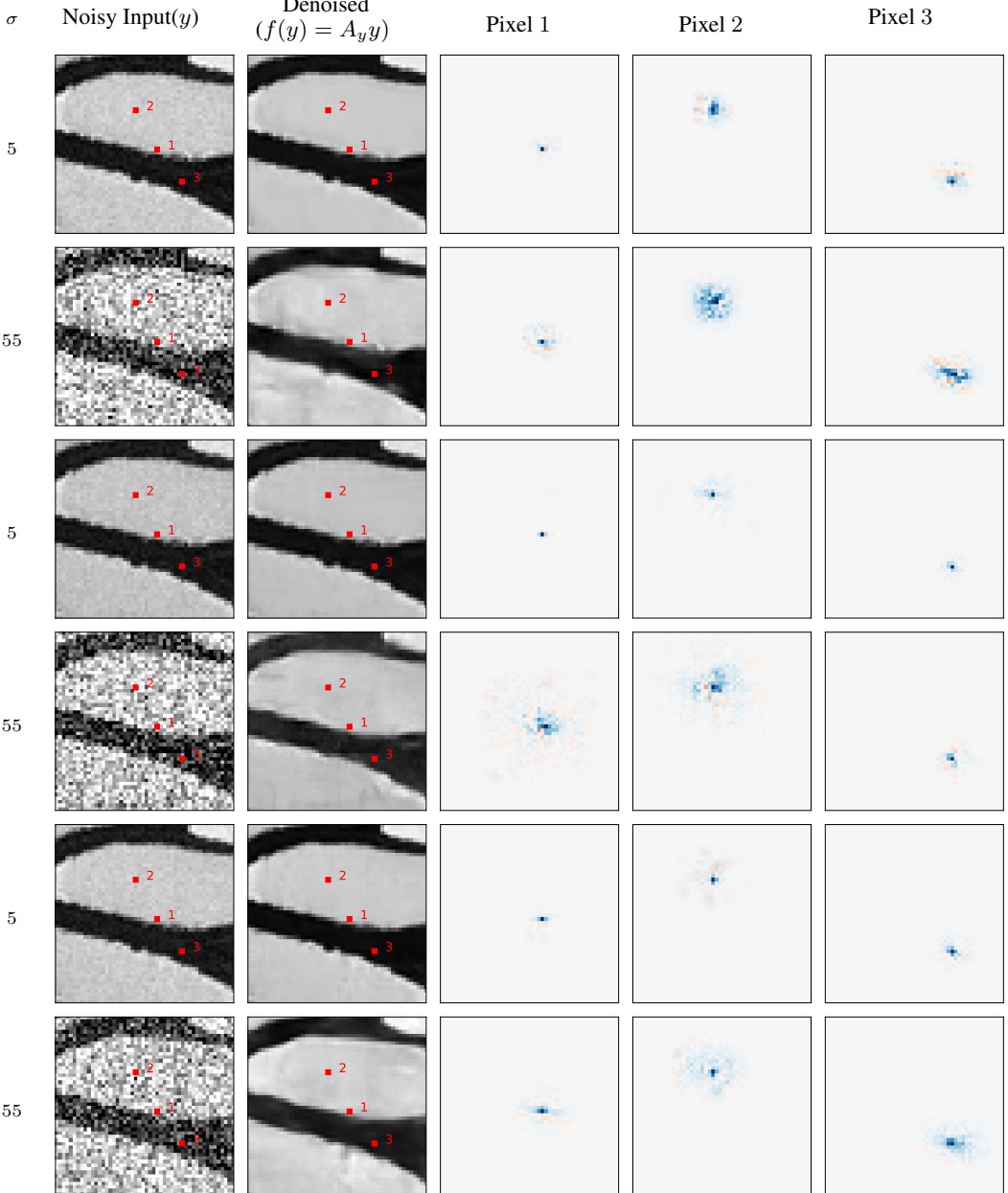

Figure 15: Visualization of the linear weighting functions (rows of $A_y$) of Bias-Free Recurrent-CNN (top 2 rows) (Section A.2), Bias-Free UNet (next 2 rows) (Section A.3) and Bias-Free DenseNet (bottom 2 rows) (Section A.4) for three example pixels of a noisy input image (left). The next image is the denoised output. The three images on the right show the linear weighting functions corresponding to each of the indicated pixels (red squares). All weighting functions sum to one, and thus compute a local average (although some weights are negative, indicated in red). Their shapes vary substantially, and are adapted to the underlying image content. Each row corresponds to a noisy input with increasing $\sigma$ and the filters adapt by averaging over a larger region.

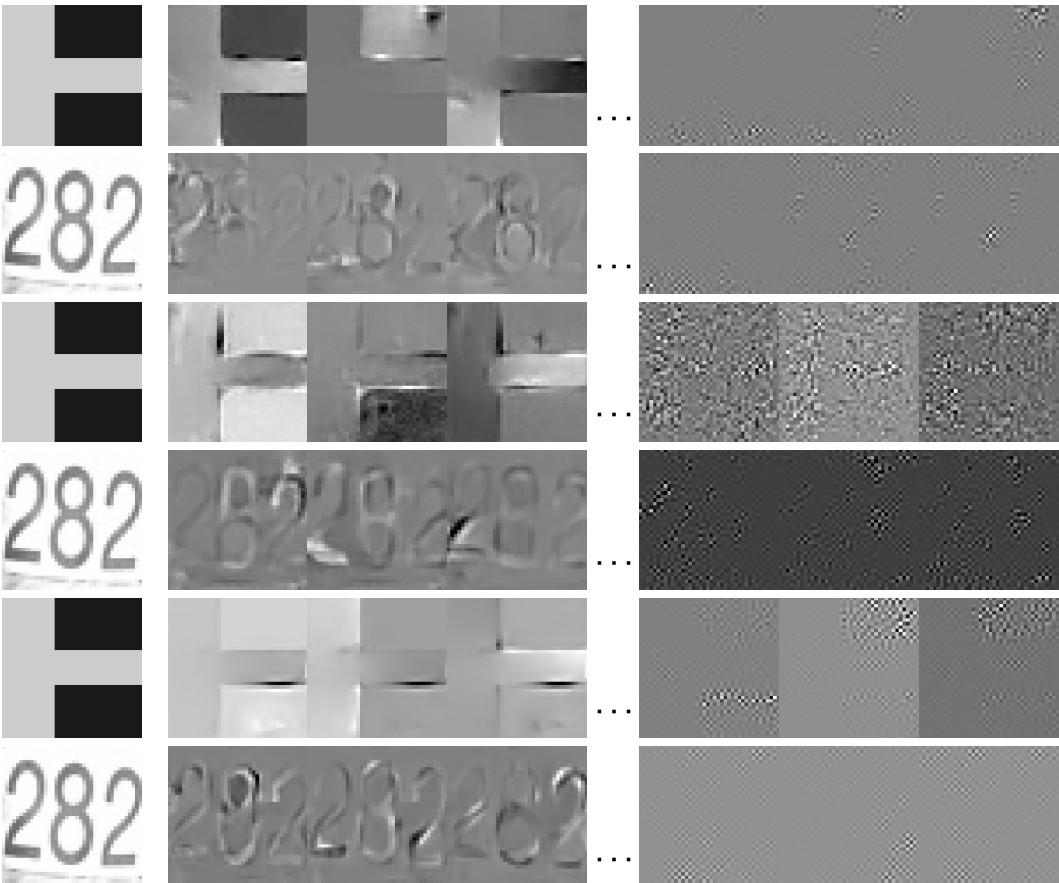

Figure 16: Visualization of left singular vectors of the Jacobian of a BF Recurrent CNN (top 2 rows), BF UNet (next 2 rows) and BF DenseNet (bottom 2 rows) evaluated on three different images, corrupted by noise with standard deviation $\sigma = 25$. The left column shows original (clean) images. The next three columns show singular vectors corresponding to non-negligible singular values. The vectors capture features from the clean image. The last three columns on the right show singular vectors corresponding to singular values that are almost equal to zero. These vectors are noisy and unstructured.

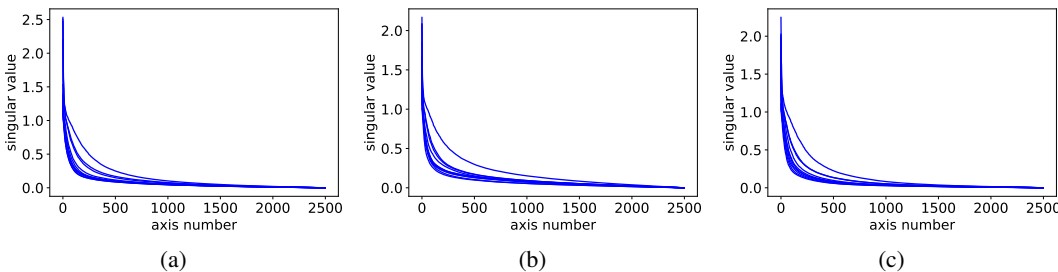

          (a)                                 (b)                                 (c)

Figure 17: Analysis of the SVD of the Jacobian of BF-CNN for ten natural images, corrupted by noise of standard deviation $\sigma = 50$. For all images, a large proportion of the singular values are near zero, indicating (approximately) a projection onto a subspace (the *signal subspace*). **(a)** Recurrent architecture inspired by DURR (Zhang et al., 2018a). **(b)** Multiscale architecture inspired by the UNet (Ronneberger et al., 2015). **(c)** Architecture with multiple skip connections inspired by the DenseNet (Huang et al., 2017).

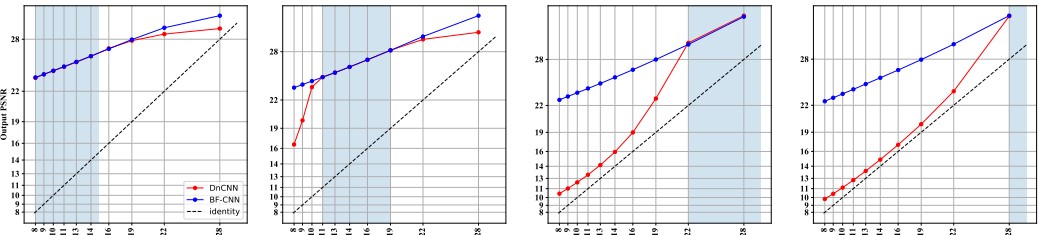

Figure 18: Comparison of the performance of a CNN and a BF-CNN with the same architecture for the experimental design described in Section 5. The networks are trained using i.i.d. Gaussian noise but evaluated on noise drawn i.i.d. from a uniform distribution with mean 0. The performance is quantified by the PSNR of the denoised image as a function of the input PSNR of the noisy image. All the architectures with bias perform poorly out of their training range, whereas the bias-free versions all achieve excellent generalization across noise levels, i.e. they are able to generalize across the two different noise distributions. The CNN used for this example is DnCNN (Zhang et al., 2017); using alternative architectures yields similar results (see Figures 19).

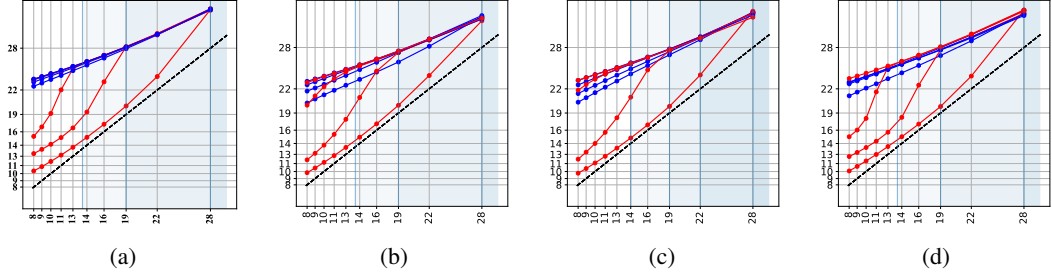

Figure 19: Comparisons of architectures with (red curves) and without (blue curves) a net bias for the experimental design described in Section 5. The networks are trained using i.i.d. Gaussian noise but evaluated on noise drawn i.i.d. from a uniform distribution with mean 0. The performance is quantified by the PSNR of the denoised image as a function of the input PSNR of the noisy image. All the architectures with bias perform poorly out of their training range, whereas the bias-free versions all achieve excellent generalization across noise levels, i.e. they are able to generalize across the two different noise distributions. (a) Deep Convolutional Neural Network, DnCNN (Zhang et al., 2017). (b) Recurrent architecture inspired by DURR (Zhang et al., 2018a). (c) Multiscale architecture inspired by the UNet (Ronneberger et al., 2015). (d) Architecture with multiple skip connections inspired by the DenseNet (Huang et al., 2017).

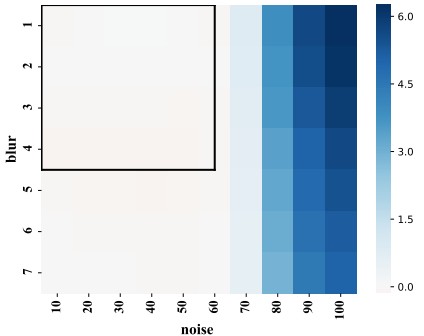 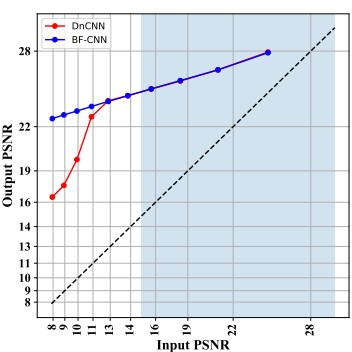

Figure 20: Comparison of the performance of DnCNN and a corresponding BF-CNN for image restoration. Training is carried out on data corrupted with Gaussian noise $\sigma_{\text{noise}} \in [0, 55]$ and Gaussian blur $\sigma_{\text{blur}} \in [0, 4]$. Performance is measured on test data for inside and outside the training ranges. **Left:** The difference in performance measured in $\Delta\text{PSNR} = \text{PSNR}_{\text{BF-CNN}} - \text{PSNR}_{\text{DnCNN}}$. The training region is illustrated by the rectangular boundary. Bias-free network generalizes across noise levels for each fixed blur levels, whereas DnCNN does not. However, BF-CNN does not generalize across blur levels. **Right:** A horizontal slice of the left plot for a fixed blur level of $\sigma_{\text{blur}} = 2$. BF-CNN generalizes robustly beyond the training range, while the performance of DnCNN degrades significantly.

