# OpenReview forum: "Robust And Interpretable Blind Image Denoising Via Bias-Free Convolutional Neural Networks"
_ICLR.cc/2020/Conference — Accept (Poster)_

### Official Review · AnonReviewer2 · 2019-10-23
**Official Blind Review #2**

**Rating:** 6

**Review:**

This paper proposed to remove all bias terms in denoising networks to avoid overfitting when different noise levels exist. With analysis, the paper concludes that the dimensions of subspaces of image features are adaptively changing according to the noise level. An interesting result is that the MSE is proportional to sigma instead of sigma^2 when using bias-free networks, which provides some theoretical evidence of advantage of using BF-CNN.

One main practical concern is that only Gaussian noise is considered in this paper which provides good theoretical analysis. It would be interesting to see if this BF-CNN is extendable to more noise types.

One unclear statement is that for UNet, the Lemma 1 seems to be no longer valid as skip connections exist. Can you provide additional proof that the scale invariance still holds in this case?



**Experience Assessment:**

I have read many papers in this area.

**Review Assessment: Checking Correctness Of Derivations And Theory:**

I assessed the sensibility of the derivations and theory.

**Review Assessment: Checking Correctness Of Experiments:**

I assessed the sensibility of the experiments.

**Review Assessment: Thoroughness In Paper Reading:**

I read the paper at least twice and used my best judgement in assessing the paper.

---

> ### Author Response · Authors · 2019-11-14
> **Response to Reviewer #1**
>
> Comment 1: "One main practical concern is that only Gaussian noise is considered in this paper which provides good theoretical analysis. It would be interesting to see if this BF-CNN is extendable to more noise types."
>
> This an interesting point. We have trained bias-free networks on uniform noise and find that they generalize outside the training range. Furthermore, we have also verified that our networks trained for Gaussian noise can generalize well when tested on uniform noise. We have updated our discussion section to communicate these results as well.
>
>
> Comment 2: "One unclear statement is that for UNet, the Lemma 1 seems to be no longer valid as skip connections exist. Can you provide additional proof that the scale invariance still holds in this case?"
>
> This is a good point. The proof of Lemma 1 is stated for a feedforward network, but the DenseNet and UNet architectures both have skip connections to intermediate layers where feature maps are concatenated. However, the concatenation operation is linear. Therefore, the entire cascade still only consists of linear units and ReLUs. For a fixed input, this means that the whole cascade is scale invariant.  We have updated the paper to address this.

---

### Official Review · AnonReviewer1 · 2019-10-23
**Official Blind Review #1**

**Rating:** 6

**Review:**

This paper studies the generalization properties of convolutional neural networks for image denoising. This paper shows that removing constant terms from CNN models provides strong generalization across noise levels. Also, this paper provides the interpretability of the denoising model based on a linear-algebra method.

Q1. Is there the possibility that the model without the constant term shows strong generalization in other image processing tasks such as image deblurring and dehazing? It would be better to discuss this point to make the scope of the proposed method clear.

Q2. About the visualization of the linear weighting functions in Figure 4, what can we figure out from the visualization, especially for image denoising? If possible, please elaborate on it.

**Experience Assessment:**

I have published one or two papers in this area.

**Review Assessment: Checking Correctness Of Derivations And Theory:**

I did not assess the derivations or theory.

**Review Assessment: Checking Correctness Of Experiments:**

I assessed the sensibility of the experiments.

**Review Assessment: Thoroughness In Paper Reading:**

I read the paper at least twice and used my best judgement in assessing the paper.

---

> ### Author Response · Authors · 2019-11-14
> **Response to Reviewer #2**
>
> Q1." Is there the possibility that the model without the constant term shows strong generalization in other image processing tasks such as image deblurring and dehazing? It would be better to discuss this point to make the scope of the proposed method clear."
>
> We agree with the reviewer that this is an interesting point. We have applied our methodology to image restoration (simultaneous deblurring and denoising). Preliminary results indicate that bias-free networks generalize across noise levels for a fixed blur level, whereas networks with bias do not. An additional question is whether it is possible to achieve generalization across blur levels. Our initial results indicate that removing bias is not sufficient to achieve this (a possible explanation is that additional structure beyond scaling invariance may be needed). We have updated our discussion and future work section to address this.
>
>
> Q2. "About the visualization of the linear weighting functions in Figure 4, what can we figure out from the visualization, especially for image denoising? If possible, please elaborate on it."
>
> For any given input image, the network operates by computing a linear function of the input. Thus, each output (denoised) pixel is computed as a weighted sum of input pixels.  The last three columns of figure 4 visualize these weighting functions (filters). We see that these filters adapt to the edges and other structures in the image, averaging over pixels that corresponding to nearby “similar” content, and avoiding content that is far away, or very different.  For example, pixel 3 is on a tree branch, and denoised by averaging over a collection of nearby pixels lying along the branch, but not including pixels from the background (similar examples can be seen in supplementary Figs 13, 14 and 15). The weighting function for pixel 1 is completely different, in both shape and size. Thus, even though the network architecture is translation-invariant (i.e., consisting of convolutions and nonlinear point operations) it exhibits spatially adaptive behaviors. In addition to this, one can also see that the effective area over which the network averages goes up with increasing noise level.  This is to be expected for denoising methods (e.g., the classical linear solution - Wiener filter - also does this). We have updated our manuscript to clarify these points.

---

### Official Review · AnonReviewer3 · 2019-10-29
**Official Blind Review #3**

**Rating:** 6

**Review:**

This paper looks at how deep convolutional neural networks for image denoising can generalize across various noise levels. First, they argue that state-of-the-art denoising networks perform poorly outside of the training noise range. The authors empirically show that as denoising performance degrades on unseen noise levels, the network residual for a specific input is being increasingly dominated by the network bias (as opposed to the purely linear Jacobian term). Therefore, they propose using bias-free convolutional neural networks for better generalization performance in image denoising. Their experimental results show that bias-free denoisers significantly outperform their original counter-parts on unseen noise levels across various popular architectures. Then, they perform a local analysis of the bias-free network around an input image that is now a strictly linear function of the input. They empirically demonstrate that the Jacobian is approximately low-rank and symmetric, therefore the effect of the denoiser can be interpreted as a nonlinear adaptive filter that projects the noisy image onto a low-dimensional signal subspace. The authors show that most of the energy of the clean image falls into the signal subspace and the effective dimensionality of this subspace is inversely proportional to the noise level.

Even though it is theoretically not too well-motivated in the paper why the bias term degrades generalization performance, the experimental results seem to clearly demonstrate the merit of bias-free denoisers. Moreover, the analysis of the network Jacobian and its interpretation as a nonlinear adaptive filter provides some interesting insight in the local properties of bias-free denoisers. Therefore, I would recommend accepting this paper, if the authors provide a theoretical discussion on why the bias term might degrade generalization performance.

Some smaller comments:
-It is not clear if d should be multiplied by \sigma^2 in its definition on page 7. The definition mentions dependence on noise variance, but the formula does not have it.
-In Section 3 in the expression of the mean squared error it is not defined what g(y) means.
-Axis labels are missing on Fig. 3.

**Experience Assessment:**

I have read many papers in this area.

**Review Assessment: Checking Correctness Of Derivations And Theory:**

N/A

**Review Assessment: Checking Correctness Of Experiments:**

I assessed the sensibility of the experiments.

**Review Assessment: Thoroughness In Paper Reading:**

I read the paper at least twice and used my best judgement in assessing the paper.

---

> ### Author Response · Authors · 2019-11-14
> **Response to Reviewer #3**
>
> Comment 1: "Even though it is theoretically not too well-motivated in the paper why the bias term degrades generalization performance, the experimental results seem to clearly demonstrate the merit of bias-free denoisers. Moreover, the analysis of the network Jacobian and its interpretation as a nonlinear adaptive filter provides some interesting insight in the local properties of bias-free denoisers. Therefore, I would recommend accepting this paper, if the authors provide a theoretical discussion on why the bias term might degrade generalization performance."
>
> We agree with the reviewer that a theoretical discussion would be a useful addition to the paper. We have added such a discussion in the first paragraph of Section 7. To recap, in this work, we show that removing constant terms from CNN architectures ensures strong generalization across noise levels, and also provides interpretability of the denoising method via linear-algebra techniques. We provide insights into the relationship between bias and generalization through a set of observations. Theoretically, we argue that if the denoising network operates by projecting the noisy observation onto a linear space of “clean” images, then that space should include all rescalings of those images, and thus, the origin. Networks with bias do not have this property, whereas bias-free networks do. Empirically, in networks that allow bias, the net bias of the trained network is quite small within the training range. However, outside the training range the net bias grows dramatically resulting in poor performance, which suggests that the bias may be the cause of the failure to generalize. In addition, when we remove bias from the architecture, we preserve performance within the training range, but achieve near-perfect generalization, even to noise levels more than 10x those in the training range.
>
> Minor comments:
>
> -"It is not clear if d should be multiplied by \sigma^2 in its definition on page 7. The definition mentions dependence on noise variance, but the formula does not have it."
>
> Thank you for pointing this out. We define "effective dimensionality" as the amount of variance captured by applying the linear map to an $N$-dimensional Gaussian noise vector with variance $\sigma^2$, normalized by the noise variance. Since the dimensionality is defined as the fraction $\frac{E_n ||A_y n||^2}{\sigma^2}$,  $\sigma$ doesn’t show up in the formula. We have updated our manuscript to make this clearer.
>
> -"In Section 3 in the expression of the mean squared error it is not defined what g(y) means."
>
> The mean-squared error is minimized over the denoising function g. In deep learning, the denoising function g is parameterized by the weights of the network, so the optimization is over these parameters. We have updated the manuscript to make this clearer.
>
> -"Axis labels are missing on Fig. 3.
>
> Thank you for noticing this - we have fixed it (adding a single pair of axes that are then shared across all subfigures).

---

### Author Response · Authors · 2019-11-14
**Response to reviewers**

We thank the reviewers for their time, and for their valuable feedback, which will improve the quality and clarity of the manuscript. We have changed the manuscript to address every comment. We refer to the individual response to each comment for more details.

---

### Public Comment · ~Yujen_Chen1 · 2019-12-22
**Question for Figure 11, 12, 19**

I have some simple questions with the figure 11, 12, 19.
According to my understanding, the x-axis and the y-axis in the figures stand for the PSNR or SSIM of the input image, and  the denoised image, respectively. (Just as figure 3).
I suggest every subfigure (a) of those three figures should add an axis title to make them clear.

Moreover, in those figures, there are four red curves which stand for the network with the net bias.
What is the difference for the four red curves in a subfigure? Network trained with different noise level range?
Also, it seems that there exists a red curves having exact power with the bias-free network, what is the setting of this curve?

---

> ### Author Response · Authors · 2020-02-14
> **Re: Question for Figure 11, 12, 19**
>
> Thank you for your questions, Yujen.
>
> The x and y axis in the images correspond to input PSNR (or SSIM) and output PSNR (or SSIM) respectively. We have made this clear in the caption and omitted the axis labels to avoid repetitive labelling.
>
> Each red curve corresponds to a network trained on a different range of noise. The noise region over which the network is trained is highlighted by different shades of blue and the largest training noise level is highlighted by a vertical line in blue color. This is similar to the convention followed in figure 3.
>
> The red curve that is being referred was trained on noise levels up-to ~14 PSNR which corresponds to a standard deviation of 55 (relative to pixel values in the range of 0 to 255). As shown by figures 11, 12 and 19 the generalization capabilities of network with bias also depends on the specific architecture. The trend seen in fig 11, 12 and 19 is consistent with the trend seen in fig 8, where we do a first order analysis of the function that the network implements.

---

### Decision · Program_Chairs · 2019-12-19

**Decision:**

Accept (Poster)

**Comment:**

This paper focuses on studying neural network-based denoising methods. The paper makes the interesting observation that most existing denoising approaches have a tendency to overfit to knowledge of the noise level. The authors claim that simply removing the bias on the network parameters enables a variety of improvements in this regard and provide some theoretical justification for their results. The reviewers were mostly postive but raised some concerns about generalization beyond Gaussian noise and not "being very well theoretically motivated". These concerns seem to have at least partially been alleviated during the discussion period. I agree with the reviewers. I think the paper looks at an important phenomena for denoising (role of variance parameter) and is well suited to ICLR. I recommend acceptance. I suggest that the authors continue to further improve the paper based on the reviewers' comments.